# Bio-Based and Robust Polydopamine Coated Nanocellulose/Amyloid Composite Aerogel for Fast and Wide-Spectrum Water Purification

**DOI:** 10.3390/polym13193442

**Published:** 2021-10-07

**Authors:** Maxime Sorriaux, Mathias Sorieul, Yi Chen

**Affiliations:** 1Scion, 49 Sala Street, Private Bag 3020, Rotorua 3046, New Zealand; maxime.sorriaux@sorbonne-universite.fr (M.S.); Mathias.Sorieul@scionresearch.com (M.S.); 2Physico-Chimie des Electrolytes et Nanosystèmes InterfaciauX (PHENIX), Sorbonne Université, CNRS, 75005 Paris, France

**Keywords:** nanocellulose, amyloid, polydopamine, nanofibrils, aerogel, wide range contaminant adsorption, water purification

## Abstract

Water contamination resulting from human activities leads to the deterioration of aquatic ecosystems. This restrains the access to fresh water, which is the leading cause of mortality worldwide. In this work, we developed a bio-based and water-resistant composite aerogel from renewable nanofibrils for water remediation application. The composite aerogel consists of two types of cross-linked nanofibrils. Poly(dopamine)-coated cellulose nanofibrils and amyloid protein nanofibrils are forming a double networked crosslinked via periodate oxidation. The resulting aerogel exhibits good mechanical strength and high pollutants adsorption capability. Removal of dyes (rhodamine blue, acriflavine, crystal violet, malachite green, acid fuchsin and methyl orange), organic traces (atrazine, bisphenol A, and ibuprofen) and heavy metal ions (Pb(II) and Cu(II)) from water was successfully demonstrated with the composite aerogel. More specifically, the bio-based aerogel demonstrated good adsorption efficiencies for crystal violet (93.1% in 30 min), bisphenol A (91.7% in 5 min) and Pb(II) ions (94.7% in 5 min), respectively. Furthermore, the adsorption–desorption performance of aerogel for Pb(II) ions demonstrates that the aerogel has a high reusability as maintains satisfactory removal performances. The results suggest that this type of robust and bio-based composite aerogel is a promising adsorbent to decontaminate water from a wide range of pollutants in a sustainable and efficient way.

## 1. Introduction

Access to clean water and sanitation is one of the 17 global sustainable development goals set by the UN Sustainable Development Summit [1]. However, while the world’s demand for clean water is growing, water contamination is compromising its availability [2]. The contaminants found in surface water include nutrients such as nitrogen and phosphorus [3], dyes [4], heavy metals [5] and trace organic compounds, such as pesticides, phenolic and pharmaceutical compounds [6]. Various methods, including chemical precipitation, chemical oxidation or reduction, ion exchange, membrane separation and adsorption have been developed for contaminated water purification. Among these methods, adsorption via sequestration of the contaminants in favourable sites of adsorbent materials has been used as the most promising water remediation strategy due to its high efficiency, recyclability, easy handling and low cost [6,7].

Aerogels are attractive candidates for water contaminant adsorption applications due to their large surface area, low density and ultra-porous structure. Aerogels can be made of inorganic materials, polymers and carbon-based materials [8,9,10]. A large and diverse array of bio-based nanofibrillar materials can also be used to create sustainable aerogels. Polysaccharides- and protein-based nanofibrils are ideal materials for the preparation of aerogels due to their renewability, biocompatibility, biodegradability and easy functionalisation [11,12,13,14,15].

Amyloid nanofibrils (AFs) are anisotropic colloids formed through self-assembly of protein or peptide monomers into cross-β sheets stacked via non-covalent hydrogen bonding [16]. Amyloid nanofibrils made from β-lactoglobulin have a semi-flexible, rod-shaped morphology with a size of 4 nm in width and up to a few micrometres in length [17]. Single AFs are extremely robust and rigid with a Young’s modulus of ~3.7 GPa [18]. The AFs are also heat, protease and detergent resistant [19]. Furthermore, AFs have a quaternary formation with a large amount of free amine functional groups available on their surface. Due to these unique chemical and mechanical properties, AFs recently emerged as potential aerogel materials [20,21]. However, the mechanical strength of plain AFs networks is only the results of molecular entanglement. Therefore, submerging an unmodified AFs aerogel network in water will cause its disintegration [22]. For this reason, cross-linking strategies have been applied to amyloid aerogels to improve their mechanical strengths and water resistance [23,24]. Peydayesh et al. (2020) used 1,2,3,4-buthanetetracaboxlic acid (BCTA) to cross-link amyloid nanofibrils to form a water-resistant aerogel, and the aerogel presented good adsorption capability for heavy metals, dyes and organic compounds [24]. The mechanical strength of that cross-linked amyloid aerogel was, however, still relatively weak (Young’s modulus: 62.5 kPa) and, therefore, should be improved for practical contaminants adsorption [25].

Polysaccharides, such as cellulose nanofibrils (CNFs), have been widely used to produce composite aerogel. They display good mechanical strength when cross-linked, and their properties can be tailored via surface functionalisation [15,26,27]. Polydopamine (pDA), a polymer that mimics mussel adhesive proteins has been coated onto nanocellulose to create a material with good mechanical strength and numerous functionalities [28,29,30]. Polydopamine has also been used as a cross-linker between nanocellulose and amine groups of polypeptides for film or aerogel preparation [31]. This bio-inspired adhesive, formed via dopamine self-polymerisation, represents one of the best advances in materials surface modification [32]. More importantly, pDA has numerous functional groups such as catechols, quinones amines and aromatic moieties which can be used not only as versatile platform for secondary surface-mediated reactions, but also for metal ions and organics compound remediation via electrostatics, chelation, hydrogen bonding, and π–π stacking interactions [25,33,34]. It was reported that a bacterial cellulose membrane coated with pDA could demonstrate good cyclability after adsorption of heavy metal ions (i.e., lead and cadmium), organic contaminants (i.e., rhodamine 6G, methylene blue and methyl orange) under a low vacuum pressure (0.7 bar) [25]. Tang et al. (2019) developed a pDA-modified double-network cellulose nanofibril aerogel that is mechanical compressible [33]. This aerogel had a superior adsorption capacity for methyl orange (256.9 mg/g) and Cu(II) (103.5 mg/g).

The possibility to combine the functionalities of AFs and pDA with the mechanical properties of a CNF network is attractive. Therefore, we aimed to create a double-network aerogel using pDA coated CNFs mixed with β-lactoglobulin-derived AFs. The two types of nanofibrils are cross-linked with an oxidation of the pDA. The resulting CNFs-pDA-AFs (CpA) cross-linked gel was then frozen and freeze-dried into CpA aerogel. This simple and organic solvent-free synthesis minimises the chemical and energy input required to prepare the aerogels. The water purification performances of this bio-based, robust and water-resistant composite aerogel are tested on a wide array of water contaminants. The adsorption capability and kinetics of CpA aerogels towards dyes (rhodamine blue (RB), acriflavine (AC), crystal violet (CV), malachite green (MG), acid fuchsin (FU), methyl orange (MO)), organic contaminants (atrazine, bisphenol A, and ibuprofen) and heavy metal ions (Pb(II) and Cu(II)) suggest that the CpA aerogel is a promising adsorbent for wide-spectrum, and efficient water remediation applications.

## 2. Materials and Methods

Materials used for the CpA aerogel preparation and adsorption experiments are listed in Appendix A.

### 2.1. Preparation of CpA Aerogel

CNF (1 wt%) solution was prepared by diluting the CNFs solution (3 wt%) with a Tris-HCl solution (100 mM) at pH 8.0–8.5. The solution was homogenised by gentle stirring (250 rpm) for one hour at room temperature (20 °C). Dopamine hydrochloride (80 mM) was added into the CNF solution and stirred to get homogeneous solution. The solution slowly became milky pink due to the polydopamine (pDA) formation via dopamine self-polymerisation. The solution was stirred overnight at room temperature with a continuous air flow to ensure a steady oxygen supply for the oxidation of the dopamine. Next, equal volumes of AF solution (1 wt%) and CNF-pDA were mixed, and the pH was measured to be around 7–7.5. This fast transition of the AFs from a pH of 2.5 to pH of 7.5 avoids degradation of Afs, which can occur around the isoelectric point of amyloid at a pH of 4–5 [18]. The final concentrations of the CpA aerogel are therefore 0.5% CNF and 0.5% AFs. After addition of 40 mM of sodium periodate (SP) solution and mixing, the solution was quickly poured into the stainless-steel cylindrical tube where it gelled. The mixture was kept overnight in a dark place to prevent photodegradation of the SP. Finally, the cross-linked gels were frozen at −80 °C or −196 °C (liquid nitrogen), followed by freeze-drying overnight in a freeze dryer (Labconco FreeZone 6 Litter Console Freeze Dryer, Kansas City, MO, USA) at −50 °C, under a pressure of 5 Pa.

### 2.2. Characterisation of CpA Aerogel

The morphologies of aerogels were observed by SEM (JSM 6700F, JEOL, Tokyo, Japan) at 3 kV accelerating voltage. Compressive tests were performed on an Instron high-capability 5900 series press (Instron Co., Norwood, MA, USA)) with a 500 N cell at a constant rate of 2 mm/min in triplicate. The CpA aerogels were compressed along the thickness direction up to 80% strain. Zeta potentials of pure CNF, pDA, CNF-g-pDA, AF and CpA at pH of 2.5, 7.0 and 8.5 were measured by ZEN 3600 zetasizer (Malvern Instruments, Worcestershire, UK) with a 4 mW laser (wavelength of 632.8 nm). The number of measurements per scan were determined automatically by the instrument. The FT-IR analysis was conducted by Bruker Tensor 27 IR Spectrometer (Bruker Optic GmbH, Ettlingen, Germany). The porosity of the aerogels was determined by calculation using Equation (1) [35]:(1)$$Porosity\;\left(\%\right)\;=\;\left(1-\frac{\rho_{aerogel}}{f_{Cellulose}\;\rho_{Cellulose}+f_{Amyloids}\;\rho_{Amyloids}}\right)\times 100$$
where $\rho_{aerogel}$ is the aerogel density, which is calculated by taking the mass of the aerogel (measured by a balance) and dividing it by the determined volume of the aerogel (determined by optical microscopy images taken using Leica MZ12). The $\;\rho_{Cellulose}$ and $\;\rho_{Amyloids}$ are the densities of CNF and AF, respectively, $f_{Cellulose}$ and $f_{Amyloids}$ are the weight fractions of the CNF and AF fibrils.

### 2.3. Adsorption Experiments

#### 2.3.1. Organic Trace Adsorption

Atrazine, bisphenol A, and ibuprofen were dissolved in 5 mL of Milli-Q water at 1 ppm. 40 mg of CpA aerogel was immerged into the organic solution with constantly shaken to ensure an enhanced adsorption at room temperature. The solution was sampled at 0, 5, 10, 20 and 60 min. The organic compound concentrations of sampling solutions were determined by high-performance liquid chromatography (HPLC) equipped with a Buchi PrepPure C-18 AQ columns on a Buchi Pure C-850 FlashPrep HPLC (BÜCHI Labortechnik AG, Flawil, Switzerland). All the tests were conducted in triplicate. The adsorption percentage was calculated by Equation (2)
(2)$$Adsorption\;percentage\;\left(\%\right)\;=\;\left(\frac{C_{initial}-C_{equilibrium}}{C_{initial}}\right)\times 100\;$$
where the $C_{initial}$ is the initial organic trace concentration and $C_{equilibrium}$ is the equilibrium concentration after aerogel adsorption.

#### 2.3.2. Dye Adsorption

Rhodamine blue (RB), acriflavine (AC), crystal violet (CV), malachite green (MG), acid fuchsin (AF) and methyl orange (MO) solutions were prepared separately at concentrations of 10 ppm. Amounts of 50 mg aerogel were immerged in 10 mL dye solutions and then incubated in a shaking incubator for 60 min at room temperature. Samples were taken at 0, 5, 10, 20, 30 and 60 min and transferred into a cuvette (10 × 10 × 45 mm) to measure their absorbance between 400–800 nm wavelengths. The dye adsorption spectrum was conducted on a Shimadzu UV-1800 UV-Visible spectrophotometer (Shimadzu Ltd., Kyoto, Japan). The intensity at the maximum wavelength for each dye (RB: 554 nm, CV: 590 nm, MG: 617 nm, MO: 461 nm, AC: 475 nm, FU: 546 nm) was used to calculate the dye adsorption by Equation (3):(3)$$Adsorption\;percentage\;\left(\%\right)\;=\;\left(\frac{I_{initial}-I_{equilibrium}}{I_{initial}}\right)\times 100$$
where the $I_{initial}$ and $I_{equilibrium}$ are the intensity of maximum wavelength of dye in sampling solution before and after aerogel adsorption, respectively.

#### 2.3.3. Heavy Metal Ion Adsorption

Copper sulphate (CuSO_4_) and lead nitrate (Pb(NO_3_)_2_) solutions in milli-Q water were prepared at concentrations of 50 ppm. Aliquots of 50 mg aerogel were dispersed in 50 mL of the metal salt solutions and shaken for 60 min at room temperature (20 °C) to ensure an active adsorption. At defined time intervals of 0, 5, 10, 20, 30, 45, and 60 min, 100 µL of supernatant was sampled and the concentration of copper and lead ions in the solution were determined by inductively coupled plasma mass spectrometer ((ICP-MS), NexION 300 ICP-MS (Perkin Elmer, Waltham, MA, USA). The adsorption percentage was calculated by Equation (4):(4)$$Adsorption\;percentage\;\left(\%\right)\;=\;\left(\frac{I_{initial}-I_{equilibrium}}{I_{initial}}\right)\times 100$$
where the $I_{initial}$ is the initial metal ion concentration and $I_{equilibrium}$ is the equilibrium concentration of metal ion after aerogel adsorption.

To determine the effect of solution pH on the Pb(II) ions adsorption performance, nitric acid (HNO_3_) and sodium hydroxide (NaOH) were used to adjust the pH of the solution range from 1 to 7. 20 mg of aerogel was placed in the Pb(II) ion solution (50 ppm) at different pH for 48 h for fully adsorption. The residual concentration of Pb(II) ions in the sampling solution was determined by ICP-MS. The regeneration of aerogel after adsorption of Pb(II) ions was further demonstrated by placing the aerogel (50 mg) in 50 mL of 50 ppm Pb solution for 20 h, and then washing the aerogel with milli-Q water equilibrated at pH 3 with HNO_3_. The adsorption of lead nitrate was again tested. This process was repeated three times.

#### 2.3.4. Adsorption Kinetics

The adsorption kinetics of contaminants were further evaluated with pseudo-first and second-order models as follows [7,9,36]:(5)$$Ln\left(q_{e}-q_{t}\right)\;=\;Lnq_{e}-k_{1}t$$
(6)$$\frac{t}{q_{t}}\;=\;\frac{1}{k_{2}q_{e}^{2}}+\frac{t}{q_{e}}$$
where *q_t_* and *q_e_* are the adsorption capability (mg g^−^^1^) of the adsorbent at equilibrium and at time *t*. k_1_ and k_2_ are the pseudo-first-order rate constant (g mg^−1^ min ^−1^) and pseudo-second-order rate constant (g mg^−1^ min ^−1^), respectively. The adsorption capability (mg g^−1^) was calculated using following equation:(7)$$q_{t}\;=\;\left(\frac{C_{0}-C_{t}}{m}\right)V$$
where *C_0_* and *C_t_* are the initial concertation (mg L^−1^) and adsorption concentration (mg L^−1^) at time *t*, respectively, *m* is the mass (g) of the CpA aerogel, and *V* is the volume (L) of the solution.

#### 2.3.5. Adsorption Mechanism

To further understand the adsorption mechanism of contaminants into the aerogel, the Weber and Morris intra-particle diffusion model was applied by using the following equation [37]:(8)$$q_{t}\;=\;k_{3}t^{1/2}+C$$
where *k*_3_ is the intra-particle diffusion rate constant (mg g^−1^ min^−½^), and C is the constant that represents the thickness or resistance of the boundary layer. 

## 3. Results

### 3.1. Synthesis and Characterisation of Aerogel

Figure 1 displays the synthesis steps and proposed chemical structure of the CpA aerogel. Dopamine self-polymerise into polydopamine (pDA) in an alkaline aqueous solution [31,38,39]. This synthetic system mimics the strategy used by mussel foot to adhere underwater onto any solid surfaces. In this study, the CNFs surface is coated with pDA (CNF-pDA) through self-polymerisation, where the catechol groups of dopamine are oxidised to dopamine quinone. The pDA coating provides a great number of active groups, such as quinone and/or OH groups able to react with the -NH_2_ groups of AFs. Finally, the oxidation agent, sodium periodate (SP) is used to convert the hydroxyphenyl from catechol-based species into quinones to increase the stability of the coating [40]. The use of SP ensures that all dopamine is converted into polydopamine and into quinones [41]. The resulting ortho-quinones by reacting with the amine groups of the AFs, via Schiff base reaction or Michael addition reactions, create stable cross-links [42]. Moreover, SP can even selectively oxidise uncoated cellulose and lead to the cleavage of the C2-C3 glycosidic bonds into 2,3-dialdehyde functions [43]. Those dialdehydes could potentially be grafted with amine, via similar mechanisms [44]. The covalent cross-linking between proteinaceous amines and ortho-quinone/dialdehydes creates a strong water-resistant double network [45], resulting in a CNF-pDA-AF (CpA) composite gel [46].

The cross-linked gels were then frozen and freeze dried into CpA aerogels. Freezing temperature strongly affects the structure and therefore the mechanical properties of aerogels [14,47]. The microscopic structure of the aerogels frozen at −80 °C and −196 °C (liquid nitrogen) were therefore analysed by scanning electron microscopy (SEM). Both samples display a porous structure composed of interconnected thin flakes (Figure 2a,b). At −80 °C, the relatively slow cooling led to the formation of large ice crystals in the CpA gel matrix. Therefore, the CpA aerogel produce are dense and they exhibit a thick flakes structure (Figure 2a inset). The density is 76.0 mg cm^−3^ and the porosity is 90.1% for aerogels frozen at −80 °C (Appendix A). Instant snap freezing the gels in liquid nitrogen (−196 °C) leads to formation of smaller ice crystals. The resulting aerogels have thinner flakes with numerous smaller pores compared to the aerogels obtained with a −80 °C freezing step (Figure 2b inset). The calculated density is 67.9 mg cm^−3^ and the porosity is 91.1% for aerogels frozen at −196 °C. Figure 2c shows a high magnification SEM image of a single flake of the aerogel frozen in liquid nitrogen (−196 °C). The high surface roughness observed could be due to the aggregation and strong oxidation of pDA nanoparticles [48]. To facilitate its use for water purification, the CpA aerogel needs to maintain its integrity once under water. Therefore, the water resilience property of the aerogel was investigated by immersing the aerogel in water with a moderate agitation (Figure 2d). It was observed that the aerogels frozen at −196 °C are quickly falling apart in water. On the contrary, the aerogels frozen at −80 °C maintain their shape and integrity when agitated under water. This difference can be ascribed to the mechanisms of ice crystal formation during freezing. The snap freezing leads to small ice crystals which are not modifying the fibrils distribution in the CpA gel. On the contrary, at −80 °C the slow growth of large ice crystals concentrates the fibrils within the interdendritic space [49]. Once the water is removed by sublimation, the resulting fibril scaffolds are sintered into a particle-packed walls with a denser structure. This more interconnected ultrastructure allows the denser aerogel to exhibit a better integrity once immerged in water. Therefore, all subsequent aerogels were frozen at −80 °C as they are more suitable for water contaminants removal applications. Upon the first immersion of the CpA aerogel in water a leachate was observed, it could probably be ascribed to the solubilisation of super-oxidised pDA macro-chains (Appendix A). The resulting aerogel is lightweight, and it can be supported by a dandelion flower seed (Figure 2e). The mechanical properties of the CpA aerogels frozen at −80 °C were further examined by performing axial compressive tests. A stress versus strain measurement up to 80% compression was generated and resulted in a three-steps curve (Figure 2f). The first phase, for strain values below 10%, is the elastic domain where the flake walls bend elastically. Then, for stain values between 10 and 50%, follows the plateau domain where the flake walls endure a plastic yielding. Finally, for strain values above 50% the densification phase leads to the collapse of the cell walls onto each other. Values of the maximum compressive stress range between 75 and 100 kPa. The Young’s modulus of CpA aerogel obtained from the stress–strain curve is 1.246 MPa, which is much higher than that of pure amyloid aerogels (62.5 kPa) [24]. The enhancement of the mechanical properties is due to the incorporation and intermolecular bonding of nanocellulose/amyloid network. Furthermore, the observed hysteresis loop shows that the aerogel has no elastic behaviour and that energy is dissipated during the test (Figure 2f).

To confirm the changes of functional chemical groups during the CpA aerogel preparation process, a Fourier transform infrared spectroscopy (FTIR) analysis was carried out. As shown in Figure 3a, pristine CNF shows two main peaks at 2896 and 3336 cm^−1^ that can be assigned to the stretching of C–H and –OH groups, respectively. In addition, the peak centred at 1057 cm^−1^ is associated with the C–O–C pyranose ring stretching variation of cellulose. Pristine amyloid shows peaks centred around 1229, 1534 and 1629 cm^−1^, which are attributed to amide I (C=O stretching), amide II (N–H bending), and amide III (C–N and N–H stretching), respectively [20,24]. Once the pDA-modified CNFs react with the amino groups from the AFs, the characteristic peaks of CNFs are diminished, while the ones from the amyloid are conserved. The two peaks at 1507 and 1607 cm^−1^, are assigned to primary amine N–H bending vibrations and aromatic C=C resonance vibration from the pDA structure [31,45]. As they are overlapping with the ones from AFs, they are difficult to observe. After dopamine polymerisation, the peak around 3336 cm^−1^ is weaker and broader indicating the formation of linkages between the -NH_2_ of AFs with OH–groups from pDA. A weak peak at 1722 cm^−1^ could be observed after SP oxidation, which is assigned to carbonyl/carboxyl groups.

The zeta potential value for CNF, pDA, CNF/pDA, AF and CpA suspensions at different pH are shown in Figure 3b. The zeta potential of pristine CNFs at pH 8.5 is −12.6 mV. After pDA coating of the CNFs, the zeta potential of CNF-pDA slightly decreased to −13 mV. This suggests that the surface groups of CNFs have been functionalised with pDA [50]. The decrease in zeta potential of CNF-pDA (−13 mV) mixed with the AFs (−10 mV) suggest strong interactions between CNF-pDA and AFs (−20 mV).

### 3.2. Adsorption of Trace Organic Compounds by CpA Aerogel

Organic contaminants are now ubiquitous, high concentrations are found in groundwaters close to agriculture, industrial and urban areas. To examine the organic contaminants removal efficiency of the CpA aerogel, we used atrazine, bisphenol A, and ibuprofen as typical organic contaminants for this experiment [24]. A fast adsorption for all three trace organic solutions at 1 ppm was reached within 5 min (Figure 4a). The fast adsorption is attributed to the aerogel’s abundant surface adsorption sites, allowing the organic contaminants to easily interact with them. The equilibrium adsorption efficiencies measured for atrazine, ibuprofen, and bisphenol A were 28.8, 65.8 and 76.7% of the compounds present in solution, respectively. The high adsorption percentage of bisphenol A and ibuprofen suggests that the CpA aerogel is a promising adsorbent for some trace organic contaminants.

The adsorption efficiency of trace organic compounds depends on many parameters, such as their molecular size, solubility in water, but also the types of surface functional groups of the adsorbent. The organics adsorption kinetics for atrazine, bisphenol A and ibuprofen at 1 ppm are further studied with a pseudo-first-order model and pseudo-second-order model (Figure 4b,c). The suitable kinetic model can be evaluated by the value of the correlation coefficient (R^2^). Both the R^2^ and q_e_ fit well with the pseudo-second-order model (Appendix A). This suggests that the organics adsorption process is a chemisorption process, where the sorption performance is dependent on the active sites and functional groups of the adsorbents. The results also suggest that bisphenol A and ibuprofen are more easily adsorbed by the functional groups/active sites of CpA aerogel compared to atrazine. The Weber and Morris intra-particle diffusion model was further applied to understand the adsorption mechanism of organics removal by the CpA aerogel. The plots of q_t_ versus t^1/2^ show two linear regions including the external surface adsorption and the intra-particle diffusion (Figure 4d). The first surface adsorption step was very fast due to the presence of large number of active sites on the CpA aerogel while the second adsorption step (intra-particle diffusion) was reduced due to the simultaneous reduction of organics in the solution.

### 3.3. Adsorption of Dyes by CpA Aerogel

Organic dyes, such as malachite green (MG) and methyl orange (MO) are widely used in the leather, textile and paper industries. They are also very common types of contaminants that can be found in environmental water. To test the aerogels capability to adsorb organic dyes, the CpA aerogels were placed into 10 ppm aqueous dye solutions, including both cationic dyes (MG, CV, RB and AC) and anionic dyes (AF and MO). After being adsorbed by the CpA aerogels, all the solutions turned transparent in less than one hour, implying a good adsorption capability towards the tested dyes (Figure 5a). This was backed up by the spectrophotometric data collected (Figure 5b). The CpA aerogel adsorption efficiencies of different dyes were measured intensively in the first hour, then again after 48 h. Each dye was rapidly adsorbed by the CpA aerogel. Within 10 min the adsorption capabilities reach between 50 and 80% then the values plateaued between 60% and 90% after 30 min. Since the binding mechanism of dye molecular on the aerogel is mainly electrostatic, π–π interactions and hydrogen bonding. The dye molecular adsorption capability strongly affects by the surface charge of dyes. For example, the presence of -SO_3_^−^ groups on the MO surface results in repulsive interactions between MO and the CpA aerogel. These repulsive interactions reduce the adsorption capability of the CpA aerogel towards MO.

The adsorption kinetics of the aerogel towards dyes were further analysed with pseudo-first order and pseudo-second order models and the fitting curves are shown in Appendix A. The parameters calculated from the kinetic model are shown in Appendix A and S5. The pseudo-second order yielded excellent fit with a high correlation coefficient (R^2^) for most dyes (except MG) and the q_e_ calculated through the pseudo-second order model is closer to the experimental value shown in Appendix A. This suggests that the adsorption of dyes onto CpA aerogel is best fitted by a pseudo-second order model. It therefore confirms that dyes are chemisorbed by the CpA aerogel and their adsorption rates are mainly depending on the active sites of aerogels rather than the dye concentrations. The plotted curves of q_t_ versus t^1/2^ show three linear regions (Appendix A), suggesting the dyes adsorption by CpA aerogel is a multi-step adsorption process.

### 3.4. Adsorption of Heavy Metal Ions by CpA Aerogel

The adsorption capabilities of the CpA aerogel towards Cu(II) and Pb(II) ions were also tested [37,51,52]. As seen in Figure 6a, the adsorption of both Pb(II) and Cu(II) ions in solution at 50 ppm onto the aerogel reach equilibrium after 5 min of incubation. This fast adsorption rate is attributed to the porous structure of CpA aerogel that allows efficient interactions between heavy metal ions and the aerogel active sites. Contrary to the low affinity towards Cu(II) ions, the CpA aerogel demonstrate a high affinity towards Pb(II) ions. The respective adsorption efficiency of Cu(II) ions and Pb(II) ions reach a maximum of 12.3 and 91.7%. The higher adsorption capability of Pb(II) ions might be attributed to the lower hydration energy of Pb(II) ions (1481 kJ mol^−1^) compared to the one of Cu(II) ions (2100 kJ mol^−1^), resulting in a preferential binding of Pb(II) ions by the CpA aerogels amine groups [52]. Moreover, the pH of solution often plays a crucial role in heavy metal ions adsorption/desorption process [15,23]. The effect of the initial pH on the adsorption of Pb(II) ions was further evaluated in a pH range between 2.5 and 7.0. In strong to mild acidic conditions (pH 2.5–5), the functional amine groups of CpA aerogel are protonated (NH_3_^+^). This results in a low adsorption capability of the positively charged Pb(II) ions (Figure 6b). When the solution pH increased from 5 to 7, the amine groups are deprotonated with an increasing electronegativity. The strong electrostatic attraction of Pb(II) ions towards the negatively charged CpA aerogel resulted in an increased adsorption capability of the Pb(II) ions.

The regeneration of the CpA aerogel for Pb(II) ions adsorption was further demonstrated by using acidic solution washing to flush out the adsorbed metal ions [24]. After adsorption of Pb(II) ions, the aerogel was placed within a pH 3 solution adjusted with HNO_3_. The shift from alkaline to acidic condition induces the desorption of ions, resulting in Pb(II) ions being released from the aerogel into the aqueous solution (Figure 6c). Cyclisation only moderately impact the remediation ability, as the adsorption capability went from 94.7%, to 81.0% and 82.8% for each of the three first cycles, respectively. Similarly, desorption values indicate that only a small amount of Pb(II) ions bond to the CpA aerogel remain adsorbed. During the cyclisation, only 0.7%, 16.5% and 12.2% of the adsorbed Pb(II) ions were not desorbed at the end of the first, second and third cycle, respectively. This reduction of adsorption rate might be due to the mass loss of adsorbent during regeneration process. Altogether, the high adsorption capability coupled with a minor decrease in adsorption efficiency after three regeneration cycles demonstrates that the CpA aerogel can easily be regenerated for reuse without additional energy input, increasing its cost-efficiency and sustainability.

To further understand the diffusion mechanism for metal ions adsorption, the adsorption kinetics of CpA aerogel was also studied. The focus was on Pb(II) ions due to the higher adsorption capability compared that of Cu(II) ions. Appendix A show the fitting results and the obtained adsorption parameters for CpA aerogel towards Pb(II) ions. The results are summarised in Appendix A. The R^2^ given by the pseudo-second order model is higher than that of the pseudo-first order model. This indicates that the Pb(II) ions adsorption by the CpA aerogel can be explained by the pseudo-second order model, and the adsorption process is chemisorption [15,23]. Moreover, the calculated q_e_ value (18.0342 mg/g) obtained from Equation (6) agrees with the experimental value (17.8254 mg/g), which also confirms that the adsorption process fits with pseudo-second order model.

The Weber and Morris intra-particle diffusion model was further applied to understand the lead ions adsorption mechanisms [37]. The relation plots of q_t_ versus t^½^ is shown in Appendix A. There are typically three stages of transport and diffusion during the adsorption process: (1) bulk diffusion of ions/molecules in the solution to the boundary layer of the adsorbent surfaces; (2) film diffusion of ions/molecules from the boundary into the adsorbent surface (intra-particle diffusion); and (3) equilibrium of adsorption. Here, the intra-particle diffusion fitting curve of Pb(II) ions adsorption by the CpA aerogel can be separated into two linear regions. The first one where the metal ions rapidly diffused from the solution to the adsorbent surface. This is due to the high porosity and large number of amino groups (-NH_2_) on the surface of CpA aerogel. The second phase is mainly controlled by intra-particle diffusion.

### 3.5. Performance Comparison

In the past, pDA, cellulosic and amyloid nanofibrils containing materials have been used separately or in pairs for organic contaminants, dyes and heavy metal ions removal from water [15,24,25,33]. Here, we report for the first time a successful incorporation of these three materials into one composite aerogel and the adsorption capability of a wide range of contaminates was demonstrated. The two bio-based nanofibrils create a double network and pDA allows the functionalisation of the CNFs and cross-linking of the network. The resulting material can be turned into a porous aerogel exhibiting good mechanical properties and shape stability in water. This aerogel is therefore an ideal candidate to efficiently adsorb water contaminants. We investigated the adsorption capability of the CpA aerogel for numerous dyes, heavy metal ions and trace organic compounds often contaminating water as the result of human activities. The adsorption kinetics results show that the pseudo-second order kinetic model can best describe the adsorption processes of heavy metal ions, trace organics contaminants and dyes. The adsorption kinetic results suggest that the adsorption process towards all the contaminants is a strong chemical adsorption. The heavy metal ions could bond with the aerogel, via phenol and amine groups, by cation–π or metal ion coordination interactions. For instance, the metal ions could be adsorbed on the CpA aerogel, by interacting with the aromatic cycles of the pDA, the lone pair of oxygen or nitrogen atoms of the CpA aerogel. Organic compounds could be adsorbed on CpA aerogel through π–π interactions or hydrogen bonding, due to the presence of phenol and amine groups on the CpA aerogel. The aerogel has the capability of adsorption of both cationic and anionic dyes, which can be attributed to the electrostatic and hydrogen bonding interaction between aerogel active sites and dye molecule. As shown in Table 1, the adsorption performance of CpA aerogel was compared with other reported adsorbents made from celluloses and proteins. It clearly indicates that CpA aerogel has a competitive adsorption capability with fast adsorption rates due to its large surface area and numerous functional groups. It suggests that the CpA aerogel has the potential to apply for wastewater treatment.

## 4. Conclusions

A novel CpA composite aerogel was developed in an eco-friendly manner by using nature-inspired pDA as a surface modifier for nanocellulose and cross-linking agents with amyloid nanofibrils via an oxidation process. In comparison to aerogels frozen in liquid nitrogen, aerogels frozen at −80 °C possess a relatively good mechanical and water stability. The adsorption of heavy metal ions, trace organics and dyes onto CpA aerogels were tested in aqueous solutions. For most of the tested contaminants targets, including Pb(II) ions, ibuprofen, bisphenol, MG, CV, RB, AC, AF and MO, a good and fast adsorption rate was found, while Cu(II) ions and atrazine were not adsorbing efficiently by the CpA aerogel. More importantly, the aerogel was able to adsorb both cationic and anionic dyes. The further adsorption kinetic studies show the best fit for the contaminant adsorption is pseudo-second-order model. The regeneration of aerogels for Pb(II) ions adsorption was further demonstrated with a simple acidic solution washing, and only 10% of the adsorption capability was lost after regenerating the aerogel three times. Hence, the CpA aerogel could be used as a renewable adsorbent for removal of a wide range of contaminants with potential for practical application in sustainable and distributed treatment of contaminated water.

## Figures and Tables

**Figure 1 polymers-13-03442-f001:**
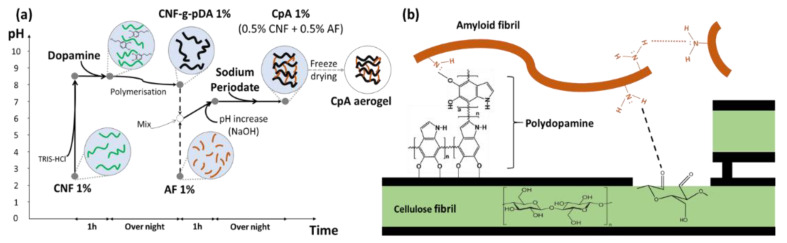
(**a**) Schematic illustration of the preparation of CNF-pDA-AF (CpA) aerogel. (**b**) Chemical interaction between CNFs, pDA and AFs.

**Figure 2 polymers-13-03442-f002:**
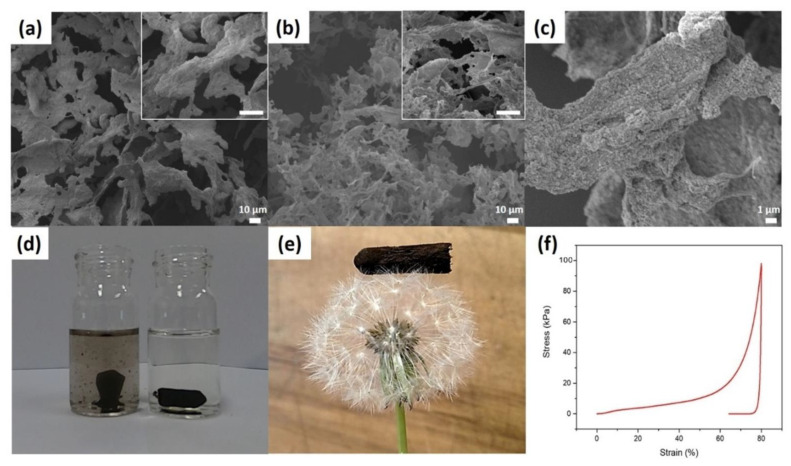
Low magnification SEM images of freeze dried CpA aerogels from freezing at different temperatures: (**a**) −80 °C; (**b**) −196 °C (liquid nitrogen), inset scale bar = 10 µm. (**c**) High magnification SEM image of CpA aerogel frozen in liquid nitrogen. (**d**) CpA aerogels in water after mild agitation frozen at −196 °C (left) and −80 °C (right). (**e**) Photograph of CpA aerogel (frozen at −80 °C) on top of dandelion seeds. (**f**) Strain-stress curve of CpA aerogel (frozen at −80 °C) under compression.

**Figure 3 polymers-13-03442-f003:**
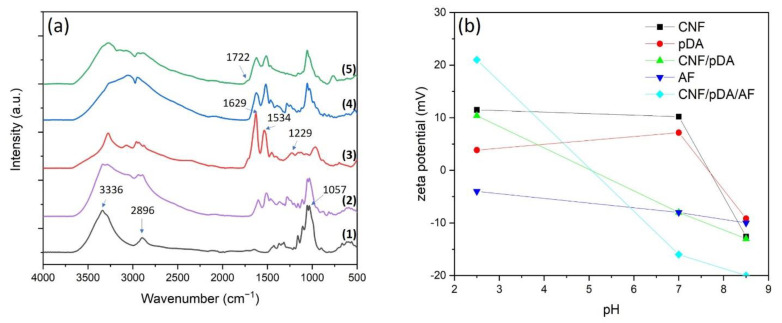
(**a**) FTIR spectra of (1) CNFs (2) CNFs-pDA (3) AFs (4) CNFs-pDA-AFs (5) CNFs-pDA-AFs after oxidation. (**b**) Zeta potential of CNF, pDA, CNF/pDA, AF and CNF/pDA/AF (CpA) suspensions at different pH.

**Figure 4 polymers-13-03442-f004:**
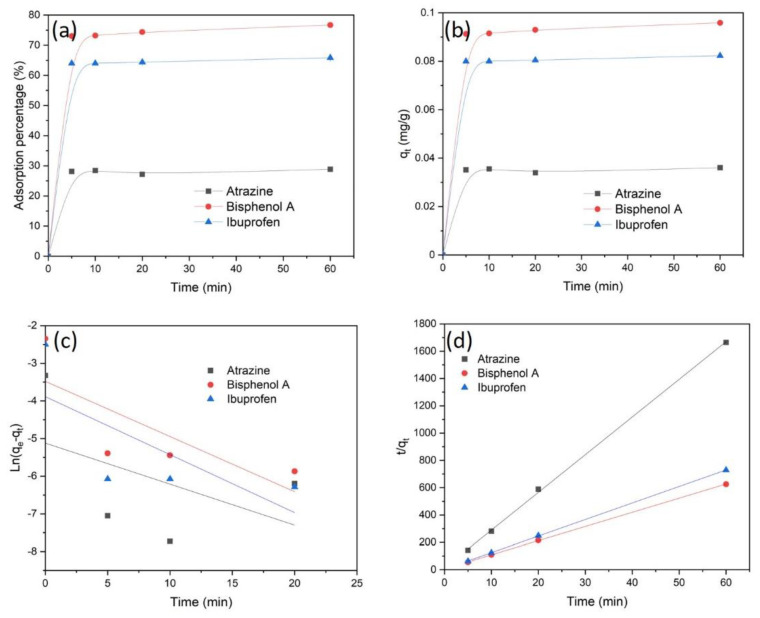
(**a**) Adsorption percentage of CpA aerogel for organic contaminants removal as a function of time. (**b**) The pseudo-first-order kinetic model fitting curve of organics on the aerogel. (**c**) The pseudo-second-order kinetic model fitting curve of organics ions on the aerogel. (**d**) Intra-particle diffusion model for organics onto the aerogel.

**Figure 5 polymers-13-03442-f005:**
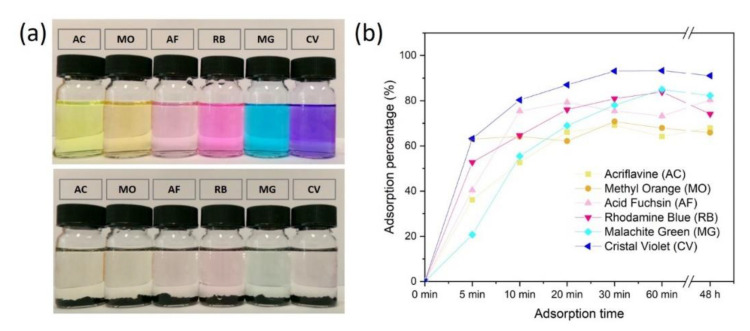
(**a**) The 10 ppm dyes solution (top) and removal of dyes by CpA aerogels after 48 h (bottom). (**b**) Dye adsorption percentage of CpA aerogel as a function of time.

**Figure 6 polymers-13-03442-f006:**
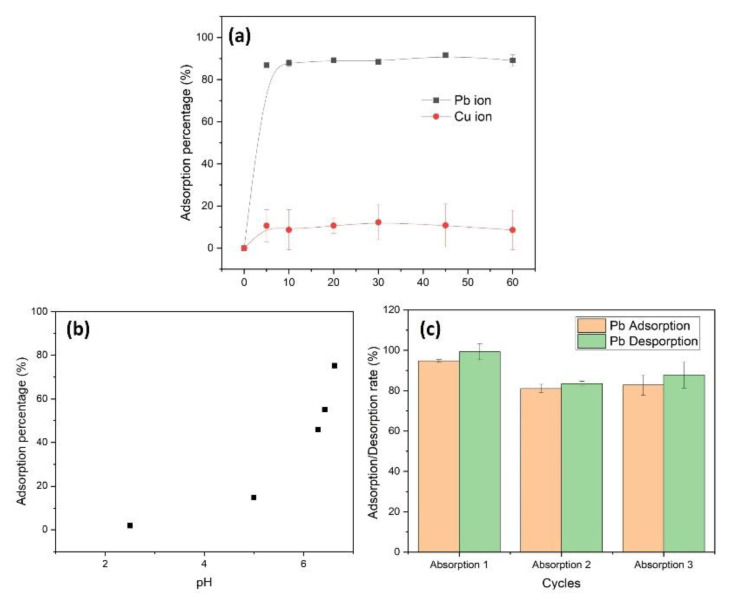
(**a**) Pb(II) and Cu(II) metal ions (50 ppm) adsorption of CpA aerogel as function of time. (**b**) Effect of solution pH on the adsorption capability of Pb(II) at 50 ppm by CpA aerogel. (**c**) Effect of regeneration on adsorption and desorption performances (%) of CpA aerogel after three cycles for Pb(II) ions (50 ppm).

**Table 1 polymers-13-03442-t001:** Comparison of relevant adsorbents reported in the literature.

Absorbent	Compound	Adsorption Capacity (mg/g)	Adsorption Efficiency (%)	Contact Time (min)	Ref.
CpA	Crystal Violet	18.66 (10 ppm)	93.1	30 min	This work
CpA	Bisphenol A	0.0958 (1 ppm)	72.9	5 min	This work
CpA	Pb (II) ion	17.8254 (20 ppm)	91.7	5 min	This work
Amyloid	Bisphenol A	0.0968 (1 ppm)	78	>60 min	[24]
Porous chitosan	Pb (II) ion	29.1 (50 ppm)	80.83	40 min	[53]
cellulose/active carbon	Methylene blue	26.7 (1000 ppm)	89	50 h	[54]
Chitosan/nanocellulose	Pb (II) ion	252.6 (150 ppm)	82	5 min	[15]

## Data Availability

The data presented in this study are available on request.

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
