# Peer review of "Bio-Based and Robust Polydopamine Coated Nanocellulose/Amyloid Composite Aerogel for Fast and Wide-Spectrum Water Purification"

_polymers, 2021, doi:10.3390/polym13193442_

Round 1

Reviewer 1 Report

In this manuscript, the authors designed a bio-based and water-resistant composite aerogel using PDA and CNFs mixed with β-lactoglobulin-derived AFs for water remediation application, this CpA aerogel presented good mechanical strength and high water contaminant adsorption capability. This work will be helpful for materials scientists to develop new water purification materials. I’d like to recommend the manuscript for publication after the following comments are carefully addressed:

  1. Could you do more characterization and discussion about the structure of the CpA aerogel and its strong chemical adsorption mechanism for water contaminants. Due to the introduction of pDA, in addition to hydrogen bonding and electrostatic interactions, the interaction between aerogel and water contaminants may also include polyphenol-metal coordination, cation-π and π-π interactions, further discussion and analysis is required. Besides, on page 12, line 365, “The heavy metal ions and … could be adsorbed on CpA aerogel through hydrogen bonding…”, this is wrong, hydrogen bonding cannot be formed by metal ions.
  2. Why did the aerogels frozen at -80 ℃ exhibit better water stability than the aerogels frozen at -196 ℃? Further discussion is needed.
  3. In Fig. 3, the horizontal axis value of FTIR spectra should be from high wavenumber to low wavenumber, and there seems to be typo spelling error in the vertical axis title.
  4. On page 5, line 185, “After dopamine polymerisaton the is weakened …”, there seems to be a grammatical error in this sentence. In addition, on page 6, line 219, it seems to be ‘Figure 4b and c’ instead of ‘Figure 4a and 4b’.
  5. In the adsorption capability-time curves, between 0-5 minutes, the adsorption capability increases rapidly, to ensure the accuracy of the curve, it might be better to take a few more points.
  6. The authors could add the following references which would again increase the interest to general functional cellulosic material readers: Journal of Bioresources and Bioproducts, 2020, 5(4): 223-237; ACS Applied Materials & Interfaces, 2021, 13, 7617-7624; Journal of Bioresources and Bioproducts, 5(2): 79–95.
  7. Also, a few very recent review papers on PDA composite material should also be included into the reference part, for example: Chem. Soc. Rev.‚ 2021, 50, 8319-8343; Chem. Soc. Rev., 2020,49, 3605-3637; Coatings 2020, 10(7), 653; Materials Horizons‚ 2021‚8, 1618-1633.

Reviewer 2 Report

This manuscript is well written and contains a lot of results. But the authors should double check the format (e.g, font size, spacing, etc) prior to submission.

Some of the comments are as follows.

I suggest the authors to arrange the figure related to zeta potential in the main text and arrange side by side with Figure 3(a) – FTIR. Also, discuss further on how the surface charge and functional groups play the role in removing heavy metal, dyes, etc.

Figure 4(a) – The symbol is TOO small and makes the readers difficult to differentiate.

Figure 5(b) – Why AC and MO showed the lowest adsorption % compared to other dyes? What are the reasons behind?

Figure 6 and 8 – Both figures can be arranged as supplementary data.

I would like to suggest authors to demonstrate the reusability of the composite aerogel for water purification. Currently, none of the results are related to reusability.

Methodology – Please double check the guidelines of the journal. Methodology should be presented prior to the results and discussion section.

Reviewer 3 Report

The manuscript entitled, ‘Bio-based and robust polydopamine coated nanocellulose/amyloid composite aerogel for fast and wide-spectrum water purification’ reported polydopamine based composite aerogel for water purification applications. The work is interesting indeed. I am mentioning some corrections before publications.

  1. Why the author used polysaccharide and protein both for water purification? What are the exact roles of these two inclusions?
  2. Is the work totally based on adsorption? What about chelation? Did the author provide some evidence about that?
  3. Is that PDA macro-chains leaches during water filtration?
  4. 7c should be modified with error bar.
  5. Did the author mention about the porosity of the aerogel? Did they consider diffusion here? If not, then provide some good arguments.
  6. Some articles based on these PDA based system are recommended for betterment of the literature review; https://doi.org/10.1021/acsami.1c08111; https://doi.org/10.1007/s11095-013-1039-y; https://doi.org/10.1039/C6RA24153K.  

Round 2

Reviewer 3 Report

Can be published